# Inter-Versus Intra-Host Sequence Diversity of pH1N1 and Associated Clinical Outcomes

**DOI:** 10.3390/microorganisms8010133

**Published:** 2020-01-17

**Authors:** Hebah A. Al Khatib, Muna A. Al Maslamani, Peter V. Coyle, I. Richard Thompson, Elmoubasher A. Farag, Asmaa A. Al Thani, Hadi M. Yassine

**Affiliations:** 1Life Science Division, College of Science and Engineering, Hamad Ben Khalifah University, Doha 34110; Qatar; halkhatib@qf.org.qa; 2Communicable Diseases Center, Hamad Medical Corporation, Doha 3050, Qatar; MALMASLAMANI@hamad.qa; 3Virology Laboratory, Hamad Medical Corporation, Doha 3050, Qatar; PCoyle@hamad.qa; 4Qatar Biomedical Research Institute, Hamad Bin Khalifa University, Doha 5825, Qatar; ithompson@hbku.edu.qa; 5Public Health Department, Ministry of Public Health, Doha 42, Qatar; eabdfarag@moph.gov.qa; 6Biomedical Research Center, Qatar University, Doha 2713, Qatar; aaja@qu.edu.qa

**Keywords:** deep sequencing technology, virus diversity, intra-host diversity, inter-host diversity, low-frequency variant, haplotype reconstruction.

## Abstract

The diversity of RNA viruses dictates their evolution in a particular host, community or environment. Here, we reported within- and between-host pH1N1virus diversity at consensus and sub-consensus levels over a three-year period (2015–2017) and its implications on disease severity. A total of 90 nasal samples positive for the pH1N1 virus were deep-sequenced and analyzed to detect low-frequency variants (LFVs) and haplotypes. Parallel evolution of LFVs was seen in the hemagglutinin (HA) gene across three scales: among patients (33%), across years (22%), and at global scale. Remarkably, investigating the emergence of LFVs at the consensus level demonstrated that within-host virus evolution recapitulates evolutionary dynamics seen at the global scale. Analysis of virus diversity at the HA haplotype level revealed the clustering of low-frequency haplotypes from early 2015 with dominant strains of 2016, indicating rapid haplotype evolution. Haplotype sharing was also noticed in all years, strongly suggesting haplotype transmission among patients infected during a specific influenza season. Finally, more than half of patients with severe symptoms harbored a larger number of haplotypes, mostly in patients under the age of five. Therefore, patient age, haplotype diversity, and the presence of certain LFVs should be considered when interpreting illness severity. In addition to its importance in understanding virus evolution, sub-consensus virus diversity together with whole genome sequencing is essential to explain variabilities in clinical outcomes that cannot be explained by either analysis alone.

## 1. Introduction 

In April 2009, an H1N1 triple-reassortant swine-origin influenza virus was isolated from humans suffering from severe respiratory symptoms in North America [1]. The virus later spread to more than 209 countries, resulting in about 14,711 deaths between April 2009 and January 2010 [2]. In August 2010, the H1N1 influenza pandemic was declared finished; however, the virus continued to circulate seasonally, replacing the former seasonal H1N1 viruses [3]. Influenza A viruses belong to the *Orthomyxoviridae* family and have a genome of eight single-stranded negative-sense RNA segments that encode 11 known proteins [4]. Surface glycoproteins, hemagglutinin (HA), and neuraminidase (NA) play major roles in attachment to host cell receptors and the release of progeny virions. Polymerase subunits PB2, PB1, and PA are critical for replication and transcription of viral RNAs [5,6]. Due to the low-fidelity of PB1, the replication process is associated with a relatively high mutation rate (2.3  ×  10^−5^), resulting in the huge genetic diversity often referred to as quasispecies [5,7,8]. 

Considering the genome diversity of the influenza virus, investigating virus evolution at the consensus sequence level will not always answer questions related to the process of evolution, pathogenicity, and transmission [8,9,10]. Thus, attention has now been shifted to studying the diversity of RNA viruses at a finer-scale, which became possible with advancements of sequencing technologies. Due to the error-prone nature of enzymes involved in RNA genome replication, RNA viruses represent a typical example of rapidly evolving viruses illustrated clearly in the unique polymorphic populations detected in a single host [8]. In contrast to other RNA viruses such as HIV and the hepatitis C virus, which can cause long-term infections allowing years of evolution [11,12], influenza virus infection typically resolves in about seven days, with peak virus shedding occurring 2–4 days after infection [13]. This short infection interval provides limited time for de novo variants to emerge and therefore to generate variants at detectable frequencies [14,15,16]. The tolerance and subsequent expansion of any acquired mutation have been found to be largely dependent on its impact on virus fitness in a particular host [17]. Interestingly, only a small proportion of within-host variants spread from one individual to another [18], of which only a few become fixed in the viral population [19] [20]. Therefore, studying the influenza virus’s evolution at the “single-host” scale is a key element that provides a predictable measure of global virus evolution. Yet, the evolutionary dynamics that transform within-host variation of H1N1 influenza viruses to global genetic diversity are poorly understood [8]. On the other hand, the intra-host diversity of RNA viruses has been also shown to affect virus virulence [21,22], immune escape [23], and antiviral drug resistance [24]. The pH1N1 virus has been circulating in the human population for a decade and hence studying its quasispecies diversity at the host and population levels is important to understand the mechanisms of virus evolution, transmission, and pathogenicity in its relatively new host. 

Despite the large number of pH1N1 sequences deposited in public databases, the majority are partial HA and NA sequences, with relatively few complete genomes. Moreover, the sub-consensus diversity of the pH1N1 virus remains largely uncharacterized with only a few papers investigating pH1N1 quasispecies features [14,25]. Here, we utilized massive data generated from NGS technology to (a) Elucidate the complete genome sequence characterization of the pH1N1 virus at the consensus sequence level in a demographically diverse country over the course of a three-year period; (b) Characterize low-frequency variants in HA and NA genes; (c) Assess inter- and intra-host virus diversity to characterize the evolutionary dynamics of pH1N1 viruses at local and global scales; and (d) Study the effect of intra- host diversity on the severity of clinical outcomes. 

## 2. Material and Methods

### 2.1. Sample Collection and Atudy Population

Nasal swabs positive for pH1N1 were collected from the virology laboratory at Hamad Medical Corporation (HMC; Doha, Qatar) during the 2015–2017 period. A total of 100 samples comprising 30 samples from 2015, 30 samples from 2016, and 40 samples from 2017 were selected for complete genome amplification. The study population belongs to more than 10 nationalities who had variable periods of residency in the country. The clinical history revealed that all patients had suffered from a fever (38 °C). Additional prominent clinical symptoms included the following: cough (62%), pharyngitis (21%), myalgia (12%), and abdominal distress (4%). Around 40% of sequenced patients were hospitalized while the rest were outpatients visiting the emergency department due to flu-like symptoms. Samples collected from patients with severe complications were collected from patients admitted to hospital after being referred from the emergency department. All patients fully recovered and none were hospitalized for more than 2 days. 

### 2.2. Ethics Statement

Samples were collected as per ethical approval # HMC/MRC-16335, issued by the HMC-IRB committee and approved by the QU-IRB committee. Clinical data analyzed were obtained from anonymized healthcare records. 

### 2.3. RNA Extraction and RT-PCR

Viral RNA was extracted directly from 1 mL of viral transport medium containing nasal swabs using a QIAamp Viral RNA Mini Kit (Qiagen; Germany) according to the manufacturer′s protocol. Quantification of the viral load (RT-qPCR) was performed using primers that target the M segment [26,27]. A standard curve relating virus copy numbers to the Ct value was generated based on 10-fold dilutions of a control sample run in triplicate. All samples were within a range of 10,000 to 25,000 genomes/µL. To amplify the coding region of the whole influenza genome, PCR amplification was carried out using a Superscript III one-step RT-PCR Platinum Taq HiFi kit (Invitrogen; CA, USA) and influenza-specific primers as described by Zhou et al., 2009 [28] to specifically enrich viral genetic material. The amplified PCR products were examined by agarose gel electrophoresis and were purified using ExoSAP-IT (Invitrogen; CA, USA) to remove excess primers and un-incorporated dNTPs. Clean PCR products were then quantified using a Qubit DNA high sensitivity assay (ThermoFisher; CA, USA) and were diluted to 0.2 ng/µL as recommended by the Illumina Nextera XT library preparation kit prior to the library preparation step.

### 2.4. Library Preparation and Sequencing

DNA libraries were constructed using a Nextera XT DNA library preparation kit (Illumina; CA, USA) according to the manufacturer’s instructions. Briefly, 5 µl of the diluted PCR sample was simultaneously fragmented and tagged with adapters by transposase provided in the library preparation kit. Afterwards, these fragments were amplified by a 12-cycle PCR program to add the indexes required for barcoding and subsequent clustering in a flow cell. The tagged and amplified fragments were then purified and size-selected using Agencourt AMpure XP beads (Beckman Coulter; CA, USA). Libraries were analyzed on a high-sensitivity DNA chip on a Bioanalyzer (Agilent Technologies; CO, USA) showing a peak size of 300–1000 bp. The concentration of all samples was then normalized before loading into the sequencing flow cell. Equal nanomolar concentrations of normalized libraries were pooled and diluted 25x in hybridization buffer (HT1). The pooled libraries were heat denatured for 2 min at 96 °C before loading in a reagent cartridge. Pooled libraries were diluted to a final concentration of 12 pM with 5% spike-in of the PhiX control (Illumina; CA, USA). Next, 600 µL of diluted-denatured libraries was added to a 300-cycle reagent cartridge. Once the paired-end sequencing run was finished, reads were demultiplexed by the instrument and assigned to each sample based on barcodes, generating FASTQ files. All raw data have been deposited into the NCBI sequence read archive (BioProject accession number PRJNA554447). 

### 2.5. Data Filtering and Read Processing

Raw sequencing reads were first mapped against the human genome (hg38) to remove virus non-specific reads. The remaining reads were then checked for quality using FastQC (version 0.11.5) [29]. Next, sequencing reads were filtered to remove adaptor contamination, ambiguous bases (N), low quality reads (Phred score <30), and fragments below the length of 50 nt using Trim-Galore and FASTX tools (version 1.33 and 3, respectively) [30,31]. Ten samples for which sequencing quality and coverage were low were excluded from downstream analyses. Further analysis of processed reads including consensus sequence calling, consensus variant calling, and low-frequency variant detection (frequency <50) was carried out using two customized workflows on a CLC Genomic Workbench (CLC Bio, Qiagen v.11). For mapping purposes, reads were aligned with the pH1N1 reference genome (A/California/07/2009) (EPI-227813) using the default penalty settings of the Burrow-wheeler aligner (BWA). All reads with mapping scores of less than 30 were discarded. Consensus sequence variants (frequency >50) were called using all mapped sequencing reads that covered each site at least 100 times and had a minimum quality score of 30. For detection of low-frequency variants, CLC workflow parameters were set as suggested by Hoecke et al. 2015 [32] with minimum coverage set to 1000x and required significance to less than 0.01 [33,34]. Here, mapping of reads was performed against the consensus sequence of each sample. Other tools were also used to confirm variant calling results obtained from the CLC Genomic Workbench including BWA-MEM (v 0.7.17) [35], SAMTools (v 1.7) [36], and LoFreq tools (v 2) [37]. Final consensus sequences were constructed using VCFtools [38].

### 2.6. Variant Analysis

A minimum of 200x coverage was used to identify variants at the consensus sequence level while an average coverage of 1000x and more was required to identify minority variants (frequency < 50). A minimum variant frequency of 2% was used as the threshold while only variants with p-value at the significance level (α < 0.01) were called [34]. Low-frequency variants were then filtered using stringent quality thresholds (average mapping quality of > 30 and an average Phred score of >30). Processing-introduced errors that occur during RT-PCR and NGS sequencing can confound the identification of true low-frequency variants; hence, we used two approaches to analyze these variants independently. LoFreq is one of the available computational methods that aims to distinguish true low-frequency variants from erroneous variants. It was used in this study to identify variants at the sub-consensus level using the recommended settings for RNA viruses [37]. Each sample was re-analyzed using a low-frequency detection tool available in the CLC Genomic Workbench (Qiagen; Germany) with a significance of <0.01 for detecting variants. The required significance option in the CLC Genomic Workbench applies statistical tests to minimize the calling of false-positive variants that can result from sequence-specific errors on the Illumina platform. Only variants identified by both tools were considered valid. The validated low-frequency variant populations identified within each sample were then compared to those found in other samples. We also compared variants called at 1% and a more conservative cutoff of 2% and found no significant qualitative nor quantitatively differences. H1 and N1 numbering schemes were used for HA and NA genes, respectively. The numbering for other genes was sequential, with the number one residue assigned to the first methionine residue at the N-terminal. 

### 2.7. Mutation Rate and Phylogeny Construction

Consensus sequences were aligned using the ClustalW program implemented in BioEdit v.7.2.6 [39]. Additional sequences representing vaccine strains (A/California/2009 and A/Michigan/2015) as well as representative sequences of different clades used for the phylogenetic analysis were obtained from the NCBI Influenza Research Database (www.fludb.org) and GISAID EpiFlu (www.gisaid.org) (Appendix A). The Bayesian phylogenies of all genes were inferred using BEAST version 1.8.4 [40,41] using the HKY+G nucleotide substitution model. For the clock model, a strict molecular clock was used while a Bayesian coalescent model was used to generate the phylogenetic tree. Eight BMCMC chains were run for 100–200 million states (based on the length of the gene analyzed), with sampling every 10,000 states. The convergence of the chains was assessed using Tracer version 1.6. All generated phylogenetic trees were visualized and edited using FigTree, version 1.4.2 (http://tree.bio.ed.ac.uk/software/figtree/). Amino acid identity was calculated using online Flusurver (http://flusurver.bii.a-star.edu.sg). Phylogenetic analysis and inter and intra-population diversity were performed using the Jones Taylor Thornton (JTT) model implemented in the neighbor-joining method in MEGA7.0.26 with 1000 bootstrapping replicates to assess statistical confidence. Finally, site-specific selection pressure was estimated using ratios of non-synonymous [42] to synonymous nucleotide substitutions per site per year by the HYPHY software package, http://www.datamonkey.org [43].

### 2.8. Analysis of Low-Frequency Variants in the Shadow of Global Variation

To identify sites of global variation in pH1N1 viruses, a total of 13,021 HA sequences and 10,971 NA sequences deposited in the NCBI Influenza Research database (IRD) and GISAID EpiFlu (www.gisaid.org) from Asia, Africa, and Europe (AAE) during the period from January 2015 to April 2018 were retrieved for comparative analyses. Using Mega 7.0.26, the evolutionary distances between sequences of the same year were calculated by computing the proportion of amino acid differences between each pair of sequences. Outlier sequences showing significant deviation from other sequences isolated during the same year were excluded from comparison. 

### 2.9. Haplotype Reconstruction and Diversity Analysis

Co-Variation Mapper was used to identify correlated amino acid variants found in high prevalence among HA sequences at the sub-consensus level [44]. To achieve that, filtered reads were aligned with the HA sequence of the reference virus using more stringent alignment options in BWA allowing a maximum of 1 mismatch per aligned read. The output SAM files were then passed to the Co-VaMa scripts to find evidence of linkage disequilibrium (LD) [44]. Resulting LD mean values for each mutation pair were then evaluated, and association was confirmed for any mutation pair with LD values that exceeded the 3σ significance limit to ensure high confidence. To study quasispecies diversity, QuasiRecomb was used to reconstruct HA haplotypes from sequencing reads obtained from each sample [45]. Haplotype assembly was done for nearly full-length HA genes (18–1680 bp) of samples showing at least 1000x depth coverage of the HA gene (n = 82 samples). QuasiRecomb implements a probabilistic model, which assumes that the underlying quasispecies is derived from a limited set of sequences by mutation and/or recombination. Additionally, it estimates the number and frequency of each existing haplotype. QuasiRecomb was also used to estimate intra- host diversity at both the single site level and at the haplotype level. Diversity at the single- site level was quantified for each patient sample (intra- host) as the entropy of the mutation distribution in HA sequences extracted from overlapping reads in each sample. Intra- host diversity was also assessed at the haplotype level in which both the number of assembled haplotypes and their frequencies were estimated for each sample. Finally, to identify polymorphic regions among all HA sequences at both consensus and sub-consensus levels, a total of 360 HA haplotype sequences were aligned with corresponding consensus sequences. To evaluate severity, we used the severity score system established by Capelastegui et al. (2012) to identify the severity of patients with pH1N1 infection. In brief, we stratified severity based on six factors: age >45 years, male sex, number of comorbidities, pneumonia, dyspnea, and confusion [46]. However, this system evaluates severity in adult patients, and we had pediatric patients in this study. Thus, we considered age <5 years as a risk factor as well.

### 2.10. Statistical Analysis 

For investigating the driving force that increases the odds of any identified low-frequency variant to emerge at the consensus level, we tested three parameters: variant genomic position, its prevalence among patients, and its prevalence across years using Fisher′s exact tests for categorical variables embedded in Prism7 [47]. *p*-values < 0.01 were considered statistically significant. A general linear model (GLM) was used to examine the relationship between the diversity of HA haplotypes and the severity of illness using Prism7.

## 3. Results

On average, 1,250,800 reads were obtained per sample with an average coverage of 500× for all segments. Coverage bias was detected throughout all segments with an overall higher coverage (>1000×) obtained for smaller segments (HA to NS) compared to polymerases segments. Nonetheless, even for segments with the lowest coverage depth (200×), we were able to obtain a good-quality (phred score >30) consensus sequence that had an uninterrupted open reading frame (ORF). In total, we were able to obtain the full genome sequences of 80 samples and full HA, NP, NA, M, and NS sequences of 10 samples. 

### 3.1. Molecular Characterization of pH1N1 Genes during the 2015–2017 Period

#### Phylogenetic Analysis, Mutation Rate, and Selection Pressure

Phylogenetic analysis of HA and NA genes revealed a chronological clustering of pH1N1 viruses which were found to be closely related to subgroup 6B.1, represented by the A/Michigan/45/2015 strain (Appendix A). In addition to S101N, S179N, and I233T variants characteristic of viruses belonging to group 6B.1, HA genes of viruses sequenced in 2017 were carrying S91R (93%), S181T (48%), and I312V variants in accordance with 2017 interim reports published by WHO (Figure 1). 

Comparative sequence analysis with the reference strain (A/California/07/2009) revealed an overall 98.2% (ranged between 96.4% and 99.6%) nucleotide sequence similarity among all genes. Unsurprisingly, nucleotide similarities were lowest for HA and NA genes, with 96.9% and 96.4% similarity, respectively. Similarly, the maximal substitution rate was reported for HA (3.23 × 10^−3^ substitution/site/year), then NA (3.19 × 10^−3^ substitution/site/year), while the lowest was reported for PB1 (1.18 × 10^−3^ substitution/site/year). Mutation analysis at the nucleotide level revealed that the majority of variants (80%) across eight genes were synonymous variants (Appendix A). Selective pressure analysis at the amino acid level revealed 13 positively selected sites: six in HA, one in NA, four in NP, and two in NS2. In HA, three of the positively selected sites were in the HA globular head (200, 201 and 203), all of which are situated in the 190-helix within the receptor binding site (RBS), and three in the HA stem domain (360, 361 and 458) (Appendix A). 

### 3.2. Mutation Analysis

An overall of 47 non-synonymous substitutions were observed across the amplified regions of the HA gene, half of which were already found in the 2017–2018 vaccine strain (A/Michigan/45/2015). The majority of the identified substitutions (62%) were located in the HA1 subunit: two in the receptor-binding domain (RBS; S202 and A169), and eight in the antigenic sites. Of these, four were located in the Sa site (L178I, S179N, K180Q and S181T), one in the Sb site (S202T), three in the Ca site (A156S, S220T and R222K), and one in the Cb site (S91R) (Figure 1). All HA-analyzed sequences (*n* = 90) shared 13 (out of 47) amino acid substitutions (Figure 1). Several variants also appeared sporadically, either in specific patients and/or in a specific period, reporting a prevalence of 10% or lower (Figure 1). Most of the identified substitutions were reported at the global level except for D31A, which was not reported before and was found in 2% of the viruses in this study. Other variants including L541M and K511T/E were reported as rare variants (<0.02%) in North America, but each appeared in 15% of the viruses sequenced here. Viruses sequenced in early 2016 had the lowest number of variants which could partially explain the lower number of positive cases reported in that specific year [48]. In the next season (2016–2017), on the other hand, viruses exhibited an additional 17 amino acid changes in HA compared to previous years, with frequencies ranging from 3% for nine of them, up to 100% prevalence for the other eight (Figure 1). Of these variants, group 6B.1 characteristic variants S91R and I312V were fixed in the virus population shortly after their first appearance in 2017, suggesting that these might be adaptive variants that were selected very quickly. A third mutation, S181T, also emerged in relatively high prevalence (56%) among viruses by the end of 2017. 

The other influenza surface glycoprotein, NA, exhibited 48 amino acid substitutions compared to the reference strain, 12 of which were common among all N1 sequences, in addition to 36 sporadic substitutions throughout all years (2015–2017) (Figure 1). Amino acid residues in the functional site and in the framework were conserved among all 90 viruses. In the 2015–2016 season, 20 substitutions were reported in the NA gene; however, none had a prevalence of more than 10% and none were found in subsequent years (Figure 1). Also, twelve of the identified substitutions appeared first in viruses circulated during 2016–2017 and reached a prevalence ranging from 2% to 97% in that season (Figure 1). Of note, all viruses possessed H275, indicating a retained sensitivity to the neuraminidase inhibitor, Oseltamivir. Furthermore, NA sequences were screened for known variants of reduced susceptibility to other NA inhibitors such as Zanamivir [49]. Of those, I117T and N222H (located in the framework) were reported from five patients in 2015–2016, none of which received antivirals during the course of infection. 

Polymerases, on the other hand, exhibited a limited number of variants with only a few reaching high prevalence. All amino acid sequences of the PA gene were carrying four amino acid substitutions (V100I, N204S, R221Q and L229S) in the PA-X protein and five amino acid substitutions (V100I, P224S, N321K, I330V, and R362K) in the PA protein (Figure 1) compared to the reference strain. Season-specific variants were also reported in single patients throughout of the study: 13 variants in 2015–2016 and twelve in 2016–2017 (Figure 1). In PB1, a total of 23 amino acid substitutions were reported with only three variants being fixed in all sequenced viruses (n = 80; G154D, I397M, and I435T), while the rest were sporadically reported without any specific pattern (Figure 1). The largest of the polymerases, PB2, possessed nine fixed variants: R54K, M66I, D195N, R293K, R299K, V344M, I354L, S453T, and V731I. Of these, I354 and S453 lie in the cap-binding domain that is essential for the cap-snatching mechanism during the viral RNA transcription process [50] (Figure 1). Most of the sporadically detected variants in polymerases clustered in one patient suffering from enhanced respiratory symptoms including peribronchial markings and mucus accumulation in lungs.

The matrix protein was among the most conserved genes with only five fixed variants, four in M1: V80I, M192V, Q280K, and K230R as well as the D21G mutation in M2 (Figure 1). As expected, all analyzed viruses had the M2-channel blocker (Amantadine) resistance genetic marker (S31N) in the M2 gene [51]. On the other hand, the non-structural gene (NS) was among the top three heterogonous genes with a mutation rate of 2.25 × 10^−3^ substitution/site/year. A total of six amino acid substitutions were observed in all NS1 gene analyzed (Figure 1). Sporadic variants also appeared, none of which reached a prevalence of more than 9% or were fixed in the pH1N1 virus population. With regard to NS2, three amino acid variants were detected in all viruses: N29S, T48A, and M83I (Figure 1). Analysis of virus diversity at the consensus sequence level showed similar evolution patterns to those reported worldwide with a few exceptions in the HA gene [52].

### 3.3. Beyond the Consensus Sequence: Genetic Diversity of pH1N1 at the Sub-Consensus Level

Most data available on influenza virus diversity has been derived mainly from analysis at the consensus sequence level for the HA and NA genes. Despite the recent shift of interest to studying diversity at the sub-consensus level, little is known about influenza virus diversity at the sub-consensus level and much less about within- and between host diversity [14,16,53]. Here, we compared non-synonymous low-frequency variants (less than 50% frequency) across samples throughout the study period. We only considered low-frequency variants (LFVs) that were detected in more than 2% of the reads in each sample. Overall, the average number of within-host LFVs ranged from 15 variants (SD = 13.7) in 2017 to 25 variants in 2015 (SD = 24) and 2016 (SD = 17) (Figure 2a). There were nine samples with greater than 50 low-frequency single nucleotide variants (Figure 2a). Across analyzed sequences, we identified 229 sites in the pH1N1 genome where variants arose in parallel in two or more patients: 38 in HA, 30 in each of the NA and NS genes, 17 in each of the NP and M genes, and the rest in polymerases. In all genes, the variant frequency spectrum had a bimodal distribution (Appendix A), with the majority of variants spotted either at low frequency (2–10%) or high frequency (80–100%). Despite the emergence of LFVs in all genes, they were most commonly seen in the surface glycoproteins (Figure 2b). In HA particularly, parallel emergence of LFVs arose independently in several patients, some of which reached a high frequency within the same year or in later years, while the majority of variants in other genes occurred once (Figure 2c). More importantly, some of the observed non-synonymous LFVs in the HA and NA genes arose at sites that might affect the antigenicity of HA [54] and the antiviral sensitivity of NA [55]. Therefore, we focused mainly on LFVs identified in HA and NA in subsequent analyses. 

A total of 121 LFVs were reported in HA sequences throughout the study period. Of those, 29 variants were located in the head domain, with 13 of them reported from at least two patients. In the stem domain, 27 (out of 92) variants were found in more than one patient (Figure 3a). Monitoring these variants over years revealed that some occurred in a specific year (88 variants), while others were recorded over years (33 variants) (Figure 3a). In subsequent analysis, we focused only on LFVs detected in multiple patients and investigated their presence and prevalence at the consensus sequence level. A total of 11 (28%) variants were found in the HA consensus sequences of pH1N1 viruses included in current study (Figure 3a). Of these, four variants originated in the head domain, while seven emerged in the stem domain. All four variants that originated from the HA head were also detected as LFVs in more than one year. In the HA stem though, the presentation of LFVs in the consensus sequence was higher for those that emerged in a specific year (7 variants) compared to those that appeared in multiple years (Figure 3a).

Remarkably, some of the LFVs that reached consensus sequences experienced a gradual increase in frequency and/or prevalence over time before emerging as variants in the HA consensus sequence (Figure 3b). D31 represents the most typical example of the gradual increase in frequency over years. It emerged as LFV (frequency < 2%) in 10% of viruses in 2015, then appeared in 8% of viruses in 2016 at a frequency of 10% before being seen in the HA consensus sequence of 5% of viruses by the end of 2017 (Figure 3b). Similarly, K511 appeared first in 2015 at frequencies ranging from 2% to 10% but was not detected at the consensus level until 2017, where it appeared in 15% of viruses (Figure 3b). A249I and L541M were observed at the sub-consensus level in 2016 in 4% of viruses and in the HA consensus sequences of 15% of each virus in 2017 (Figure 3b). The group 6B.1A characteristic mutation (S181T) was also detected as an LFV in 2015 before reaching the consensus level of 45% of viruses in 2017 (Figure 3b). Altogether, these data suggest that some emerging variants arise as LFVs before being detected at the consensus level and hence can help us predicting the next season’s variants.

In the NA segment, a total of 146 LFVs were reported, of which, 35 (24%) variants were shared among multiple patients, while the other 111 variants were reported only once during the study period (Figure 4a). In contrast to HA, the majority of across-patient variants (68%) were seen in one year, compared to 11 variants reported in a multiple years (Figure 4a). In total, 19 LFVs were found in the consensus NA sequences during the years of study, four of which were detected in multiple patients, while 15 were found in single patients (Figure 4a). Five of these variants exhibited a gradual increase in frequency over years before being seen at the consensus level, namely, V81G, Q64E, I389T, N449D, and D451G (Figure 4b). Specifically, V81G, I389T, and D451G were observed in consensus sequences one year after their first detection as LFVs, while N449D emerged first in 2015 (frequency < 2%), disappeared in 2016 and re-appeared in the NA consensus sequence in 2017 with 87% prevalence (Figure 4b). Nonetheless, none of these variants was fixed in the pH1N1 virus population, except for N449D and V81A, which were found in medium to high prevalence reaching between 62% and 92%, respectively. Notably, we also observed the emergence of six variants that are known to be associated with reduced sensitivity to neuraminidase inhibitors [55,56] (Figure 4a). Of these, three variants were located in the active site and three in the framework. Four of these variants were reported in single patients, while R152 and N295 variants were found in at least five patients. Four LFVs that are known to be associated with reduced sensitivity to NAIs were found in a single patient who did not receive oseltamivir or zanamivir treatment; however, the patient was on antibiotic treatment during the course of infection due to other minor respiratory complications. Moreover, three patients (all reported during 2015) were suffering from retrocardiac infiltration, pericardiac infiltrates, and peribronchial cuffing. It is worth noting that none of these patients was treated with any neuraminidase inhibitor. More importantly, none of these LFVs was found in the consensus NA sequences except for I117V, which was reported once during 2016. Together, these data suggest that NAI resistance mutation can emerge sporadically at the sub-consensus level without any exposure to NAI drugs and might rapidly evolve to consensus upon neuraminidase treatment, which poses a great threat to valuable patients such as those described above. 

Finally, we investigated the effect of variant position (head vs. stem), its temporal distribution (one-year vs. over years), its prevalence among patients (single patient vs. multiple patient), and its ability to emerge in the consensus sequence. None of these factors had a significant effect on determining variant existence in the consensus sequence of both HA and NA genes (Appendix A).

### 3.4. Low-Frequency Variants in Antigenic Sites of pH1N1 Glycoproteins 

Analysis of LFVs revealed variations in antigenic sites of HA and NA genes. However, none of the variants in antigenic epitopes was present at significantly higher frequencies compared to other variants in either gene. In HA, 17 amino acid changes were identified in these regions during the study period (Figure 5). The majority of these variants were located in the Sb site (11 variants) and two in each of the following sites: Sa, Cb, and Cb (Figure 5). Of these, A203 and D204 in Sb were the most prevalent at the sub-consensus level, appearing in >80% of viruses. On the other hand, only two variants were found in consensus HA sequences: L178 (2016) and S181T (2017) in Sa (Figure 5). While L178 was detected sporadically, S181T increased in frequency and prevalence over time. This mutation appeared as LFV (frequency of less than 2%) in 2015 before being detected in 56% of consensus HA sequences by the end of 2017. Overall, amino acid changes which originated as low-frequency variants in antigenic sites represented 50% (2 out of 4) of the total amino acid changes that emerged in antigenic sites of HA sequences during the study period. In NA, only five (out of a total of 144 LFVs) were in the NA antigenic site located within amino acids 141–155 [57]. These included R152 and Y155 in the active site and framework, respectively which have been previously reported to be associated with reduced sensitivity to NAIs. None of these variants, though, was reported from more than 2 patients. These data suggest that a change in any mutation frequency occurs mainly as a result of a stochastic process rather than selection pressure and rarely amplifies to a frequency greater than 10%. 

### 3.5. Abundance is not Necessarily Associated with Emergence at the Consensus Level

Analysis of LFVs in HA revealed the accumulation of highly prevalent LFVs within the two main regions in the HA gene: in the 199–210 region in the head region and in the 359–364 region in the stem (Appendix A). Surprisingly, 4 (out of 6) of these variants were found to be under positive selection as reported by the site-selection test performed earlier in this study. None of these variants, albeit found in high prevalence among patients (>90%), was able to reach a frequency higher than 30% in any specific patient or to be fixed at the global level. Thus, we looked for evidence of linkage disequilibrium (LD) between pairs of amino acids situated within these two regions separately, using the pipeline implemented in Co-Variation Mapper (CoVaMa v.0.1) [44]. The standard deviation and LD cutoff mean values of the stem region (359–364) were 1.2 × 10^−3^ and 9.2 × 10^−4^, respectively, yielding a 3σ critical value of 4.6 × 10^−3^. None of the amino acids of interest showed any significant LD values. Similarly, none of the targeted amino acids in the head region (199–210) were found to surpass the 3σ critical value of 7.4 × 10^−3^ and hence no association was determined among amino acids in these two regions separately. Together, these results indicate that none of these prevalent variants appeared simultaneously in any HA sequence obtained in this study. Therefore, factors other than variant prevalence can also contribute to defining the variant appearance at the consensus level.

### 3.6. Low-Frequency Variants of HA not NA Repeatedly Arise at the Consensus Level Globally

The majority of patients included in this study were from Asia, Africa, and Europe (AAE). Therefore, we investigated the circulation and prevalence of locally identified LFVs in all HA sequences of pH1N1 viruses (*n* = 13,021 HA sequences) reported from these three continents during the same period of our study (2015–2018). All variants that arose as LFVs and reached the consensus level locally were also sites of global virus variation (Figure 6). Two of these variants, L178 and S181T, were located in the Sa site which may reflect concordant antigenic selection of within-host variation at the global scale (Figure 6). However, the concordance between locally and AAE reported variants was not always present. Some variants were seen at a higher prevalence in our samples compared to global viruses such as T249I, K511T, and H290R which were found in 15% in each of our patients and in less than 1% in AAE, suggesting that these variants may have within-host benefits rather than being advantageous for global evolution. Overall, locally identified HA variants tended to emerge at sites that vary at the global scale despite the difference in prevalence. The observation was different in the NA gene, the target of currently used treatments (Figure 6). For example, a total of 28 LFVs were identified at the NA consensus level at the global scale, of which eight were found at the consensus sequences of local strains (Figure 6). 

### 3.7. Scaling up from Single Site Variation to Haplotype Diversity 

Here, we shifted from analyzing LFVs at the single amino acid level to analyzing haplotype diversity within and between patients. We considered the possibility that viral haplotypes can be shared among patients infected within the same time or at different time periods. To achieve this purpose, the underlying quasispecies obtained from deep sequencing data was used to estimate within-host HA haplotype diversity and distribution. Haplotype assembly was done using the conservative mode of the variation Bayesian modified expectation maximization (EM) algorithm implemented in QuasiRecomb [45]. Only samples with a coverage of more than 1000x for the HA gene (*n* = 82 samples) were included in this analysis. QuasiRecomb denotes HA sequences with at least one different nucleotide in this population as haplotypes. Overall, the diversity of haplotypes in each sample ranged between one haplotype in 17 patients up to 261 haplotypes in one patient (Figure 7a). The majority of patients (*N* = 32) harbored two haplotypes including the consensus HA sequence which represented more than 50% of sequencing reads per sample in most cases. The majority of children (>65%) under the age of 5 years exhibited more than two haplotypes compared to other age groups (Appendix A).

Overall, the highest numbers of haplotypes were recorded from 3 patients whose samples were collected in 2015 (28711, 27825, and 27618) (Figure 7b). Of those, sample 27,618 (44 haplotypes) and sample 27825 (256 haplotypes) were received from children (<5 years old) suffering from retrocardiac infiltration, tachycardia, and atelectasis. Moreover, twenty-three of the HA haplotypes found in sample 27618 were also reported in sample 28773 which was collected in the same year and was carrying 29 haplotypes (Figure 7b). Similarly, sample 28773 belonged to a 2-year-old patient who suffered from perihilar and pericardiac infiltrate during the flu infection. Interestingly, some low-frequency haplotypes exhibited by the 27618 and 28773 samples were reported as dominant haplotypes in samples isolated within the same year (2015) (Figure 7b). In 2016 and 2017, the diversity was limited in terms of the number of haplotypes and their relative frequencies in each sample except for three samples: #30404 (16 haplotypes), #30610 (26 haplotypes), and #30611 (16 haplotypes) collected in 2017 (Figure 7b). In addition to carrying the highest number of haplotypes among patients in 2017, the haplotypes of these three samples clustered together in phylogeny (Appendix A). Of note, all HA haplotypes of the 30611 sample were also found in the 30610 and 30404 samples (Figure 7b). However, sample 30611 belonged to a 67-year-old patient suffering from respiratory complications (peribronchial marking) and was on oseltamivir treatment while the other two patients were under the age of 30 with common cold symptoms (cough, fever, and pharyngitis) and were either on oseltamivir or antibiotic treatment. It is worth mentioning that samples 30610 and 30611 are family members who were diagnosed at the same time of having pH1N1 infection. The clustering of HA haplotypes from these patients signify the possibility of haplotype transmission among people in close contact. In accordance, sample 30404 was collected one month (July) after the collection of the 30610 and 30611 samples [58], yet was sharing some haplotypes which suggests the ability of low-frequency haplotypes to continue circulating along with the dominant strain and infect other people months after their first appearance. Together, these data suggest that some low-frequency haplotypes could be transmitted among people below current available surveillance thresholds.

Despite the sharing of haplotypes, severe symptoms were seen in some patients but not in others. Thus, we investigated the effect of haplotype diversity on the clinical outcome. Generally, 53% of patients reported to suffer from extreme respiratory and cardiac symptoms, such as pneumonia, bilateral lung congestion, and inflammatory airways, exhibited higher diversity with regard to haplotype abundance (> 6 haplotypes) (Figure 8a). Most of these patients (90%) were either under the age of five or over the age of 60. In contrast, the rest of the patients who suffered from severe complications (48%) were carrying one or two haplotypes and were within the >5- to <60-year age group with the exception of two patients: a 66-year-old patient and a 4-year-old child. Thus, in addition to intra host genetic variation, patient age should also be considered when interpreting the illness severity of influenza patients. Moreover, we compared the presence and prevalence of single nucleotide variants in HA haplotypes assembled from patients with severe complications and those with flu-like symptoms (Figure 8b). Notably, a total of 21 variants (non-synonymous variants = 48%) were found exclusively in haplotypes of patients suffering from severe respiratory and/or cardiac complications (Figure 8b). Besides, ten identified variants were detected at significantly higher prevalence in severe patients compared to patients with typical flu symptoms. Ten percent of these variants were located in HA antigenic sites such as 144 in Sa, 156 in Ca, and 210 in Sb. Further analysis of the impact of these variants on HA function is required to explain their exclusive appearance in patients with severe complications. 

Evolution of low-frequency variants over time and their emergence at the consensus level triggered us to investigate the possibility of this happening at the level of haplotypes; thus, we studied the phylogenetic relationship between haplotypes of different patients over years. At first glance, the phylogeny generated using assembled haplotypes (*N* = 440 HA sequences) yielded topologies concordant with those previously generated for HA consensus sequences alone (Appendix A). Viruses clustered chronologically, where 2017 viruses clustered discretely from those from 2015 and 2016 (Figure 9). Some of the low-frequency haplotypes identified in 2015 samples clustered with dominant HA sequences of 2016, suggesting that haplotypes that emerge in a specific year can dominate in the following year (Figure 9). On the other hand, we also noted the clustering of 2015 consensus sequences with low-frequency haplotypes found in the 2016 viruses (Figure 9). It is worth nothing that the clustering pattern of viruses in all years was not always patient specific; instead, was showing distinctive clustering patterns between haplotypes and consensus sequences from different patients. This was most clear for low-frequency haplotypes recorded from 2015 and 2016 that were clustering together in phylogeny. Likewise, haplotypes of different samples had more similarities compared to corresponding consensus sequences in 2017 which signifies the possibility of haplotype transmission among individuals during a limited time windows (Figure 9). 

Finally, we identified polymorphic sites in HA by mapping amino acid sequences of assembled haplotypes and generated a heat map (Appendix A). Alignment results revealed amino acid heterogeneity pattern specific to each year. Polymorphic regions were mostly spotted in viruses isolated during 2017 in both consensus and haplotype sequences. They were mainly located in three sites: 91 (Cb), 181 (Sa), and 312. Other less prominent sites were also identified in 2017 including 125, 146, 249, 290, and 511. In 2015, all polymorphic sites (156, 199, 290, 305, and 502) were seen at the sub-consensus level. On the contrary, the number and frequency of polymorphic sites in 2016 were very limited. Only one site was found to be heterogenic at the sub-consensus level during all years (residue 199 located near the Sb site); however, this was never detected in HA consensus sequences. Finally, we repeated the site-selection pressure test using assembled HA haplotypes. Selection pressure analysis of HA haplotypes revealed 22 positively selected sites, compared to six positively selected sites when analyzing consensus sequences only (as indicated before). This suggests that purifying selection removes deleterious variants before they appear at the consensus level. Notably, most of positively selected sites in the HA head were located in Sa (141, 173, 177) and Ca (157) antigenic sites (Appendix A). 

## 4. Discussion

During the nine seasons since its emergence, the pH1N1 virus has undergone significant genetic changes, resulting in the generation of eight genetic groups as of 2017 [49]. It is therefore of prime importance to study virus evolution at sub-consensus and consensus levels as that will broaden our current knowledge of virus evolutionary dynamics. In this study, we aimed at providing complete genome characterization of the circulating pH1N1 viruses collected between 2015–2017 at both consensus and sub-consensus sequence levels. A total of 90 pH1N1 positive samples were collected from individuals of different genetic backgrounds (12 nationalities) who had variable residencies and travel histories, a serious concern in a multinational country. All samples were collected from patients with clinical manifestation ranging from fever to severe respiratory symptoms such as pneumonia. Generally, pH1N1 viruses experienced global reduction in antigenic and genetic changes in 2016 which was clearly demonstrated by the low number of pH1N1 positive cases reported locally and in AAE [48]. In contrast, viruses in 2017 exhibited the highest number of variants at both consensus and sub-consensus levels; nonetheless, viruses in this year showed the lowest diversity at the single host level. 

The most prevalent amino acid changes were conservative except for a few major non-conservative changes. The highest number of non-conservative changes was seen in NA: G77R, I188T, and N449D, all of which were found at relatively high prevalence reaching 92% in the 2017 viruses. Some of these variants have been proven to afford beneficial changes in virus ability to survive in its relatively new host, in particular, in NS1 and PA genes. For example, the natural accumulation of six amino acid changes (E55K, L90I, I123V, E125D, K131E, and N205S) in NS1 has apparently restored its ability to inhibit host gene expression, including IFN expression [59,60]. Similarly, the accumulation of amino acid changes in the PA-X protein has resulted in decreased PA-X-mediated host gene expression shutoff and increased NS1-mediated inhibition [61,62]. On the other hand, most of the sporadically detected variants were found to negatively affect protein function and subsequently reduce virulence such as Y89H (2%), P162S (1%) R88H (3%), E172K (1%), and R140Q (1%) in NS1, which was also associated with the binding loss of NS1 to the p85 β subunit of PI3K or the activation of PI3K signaling which helps during virus replication, resulting in reduced virulence [63]. Interestingly, some of the variants reported here were only detected in specific regions around the world such as K511T/E and I421L in HA, which were only reported from North American countries. 

Despite the high number of research papers studying pH1N1 molecular evolution since its emergence in 2009, most of these papers were focused on consensus sequence analysis [64]. The diversity of the underlying viral population at the sub-consensus level, on the other hand, remains poorly understood, with few papers describing the diversity at the sub-consensus level [14,16,25]. In clinical virology, focus has shifted from the well-established detection of single variants in dominant strains to genome-wide haplotyping of within-host virus population diversity, as low-frequency haplotypes have been shown to affect virulence, immune escape, and drug resistance [21,24]. Furthermore, the presence of closely related viruses within a patient or a population is subjected to continuous competition and selection, and hence, plays a major role in driving the evolutionary process of RNA viruses [10,65,66]. Nevertheless, studies investigating within-host diversity of the influenza A virus have reported limited diversity with only very few LFVs being detected in more than one individual [14,15], suggesting that virus evolution is mainly happening due to stochastic processes. Here, we characterized viral evolution at the sub-consensus scale by investigating LFV prevalence, the extent of intra- and inter-host virus diversity, and the effect of intra-host diversity on the clinical outcome. Overall, the majority of LFVs detected in all genes were reported in single patients except for the HA gene, of which 33% of LFVs were found in at least 2 patients (Figure 2). 

Exploring diversity at a single-site level has been shown to be a reliable measure of global diversity, as the majority of the underlying within-host diversity can also be seen at a higher global scale [67,68]. Here, LFVs in the HA head encoded amino acid changes in antigenic sites (*n* = 17) and/or near the receptor-binding domain (*n* = 4). Only one of the six LFVs in the receptor-binding site, V169, emerged at the consensus level in 4% of the pH1N1 viruses in 2016. On the other hand, three of 17 LFVs that accumulated in antigenic sites reached the consensus sequence level: L178I and S181T in Sa and R222K in Ca. Restricted changes in RBS compared to a high number of antigenic epitopes diversities might provide additional evidence that pH1N1 has already shifted from the host-adaptation stage to immune-driven selection [69]. Surprisingly, although LFVs in antigenic sites were mainly found in the Sb site (11 variants), none was found in HA consensus sequences compared to variants in Sa and Ca sites. These data suggest that variants with the potential to encode antigenic variation are generated reproducibly in multiple individuals but do not become fixed in the viral population, even if they might offer the virus an immune-escaping advantage. Also, these findings suggest that other evolutionary factors may be hindering LFVs emergence at the consensus level. As an example, the N173K variant at the Sa site was detected at the sub-consensus level in 2% of individuals in this study. N173K has been experimentally found to confer escape from pH1N1-specific antibodies elicited by vaccination [70], but was not able to reach the consensus HA sequence or to be rescued when generated in reverse genetics as a pure population, demonstrating that additional compensatory variants should be acquired to restore fitness [71]. Hence, fixing of variants requires equilibrium between the alteration of antigenicity at the one end and maintaining replication and transmission fitness on the other end [8]. In total, only 17% of LFVs in HA emerged at the consensus level, further confirming that pH1N1 viruses are now well-adapted to their new host, and therefore, most variants are likely to be deleterious. A similar evolution pattern has been also reported when using a mouse model to investigate the evolution of the 1918 H1N1 pandemic, during which within-host diversity was much higher at the beginning of the pandemic compared to later transmissions [72]. Moreover, site-selection analysis at the HA consensus level revealed that 3 residues in the HA head are under positive selection, all of which are located in the RBS, whereas selection at the sub-consensus level revealed four positively selected sites (out of 5 in the HA head) in Sa and Ca antigenic sites and none in RBS (Appendix A). This might suggest that these LFVs act as baits to attract host immunity attention and conversely provide a less aggressive environment in which the dominant strain can thrive [73]. A further serology analysis to test the presence of antibodies specific to these reoccurring variants could support such a hypothesis.

Although LFV reproducibility in multiple patients was shown to be a good predictive parameter of their appearance at the consensus level (Appendix A), the evolutionary dynamics of LFVs sometimes diverge. This was most obvious for the most prevalent LFVs found in the HA head region, 199–210, and in the HA stem region, 359–364 (Appendix A). Despite their absence from the consensus sequence, selection pressure analysis revealed that four of these variants were found to be under positive selection: 201 and 203 sites in the 199–210 region and 360 and 361 sites in the 359–365 region of HA (Appendix A). Moreover, none of these variants was found to be linked together when performing disequilibrium linkage analysis using the Co-Variation Mapper pipeline. The location of these variants within the vicinity of the receptor-binding site [74] and in the conserved region of the stem domain could partially explain their absence from consensus sequences. Therefore, it would be interesting to see how these variants will evolve in the next years and their effect on virus replication fitness, transmissibility, and pathogenicity. 

Emergence of LFVs at the consensus level may depend on other compensatory and/or compulsory variants to maintain the integrity of the protein structure and functionality. Accordingly, we further analyzed virus genetic diversity within and between patients in terms of haplotype diversity and frequency. Phylogenetic analysis of HA haplotypes revealed interesting data regarding haplotype evolution and transmission, including (a) The clustering of low-frequency haplotypes of 2015 viruses with the consensus sequences of the 2016 viruses, indicating the possibility of these low-frequency haplotypes to dominate in a few months after their first appearance; (b) The clustering of dominant haplotypes from 2015 viruses with low-frequency haplotypes of 2016, denoting the gradual loss of haplotypes over time; and (c) The clustering of haplotypes of different patients infected during the same year, suggesting the possibility of haplotype transmission among infected individuals. Although haplotype transmission has been observed in the current study, the transmission was limited among patients infected within a specific period of time rather than over months or years. In accordance with our results, the majority of studies investigating virus transmission bottlenecks concluded that tight bottleneck dominates influenza virus transmission at the level of individual hosts [14,15]. However, an interesting study that was carried during the 2009 pandemic showed that up to 200 viral genomes could be transmitted among infected patients [16]. The “loose bottleneck” in the former study was previously reported during the pandemic in 1918 [72], which became more stringent throughout the host adaptation process. Increased quasispecies diversity has been previously linked to enhanced pathogenicity compared to that with a limited number of quasispecies [21]. In our analysis, the majority of patients (*N* = 9; 53%) suffering from complicated illness were carrying a higher number of haplotypes (more than 6 haplotypes), most of which (90%) were either less than 5 years old or more than 60 years old. Those patients were suffering from pneumonia, bilateral lung congestion, and perihilar and peribronchial marking. We also observed particular clinical manifestations that were associated with unique patterns of variants and haplotypes (Appendix A). Symptom severity could be driven by many factors that were not considered in this study including host-related factors, such as vaccination and immune status [15]. Unfortunately, we had restricted access to clinical data such as patients’ history. All patients included in our study recovered and none were hospitalized for more than 2 days.

In conclusion, pH1N1 viruses rapidly acquire de novo variants at both consensus and sub-consensus levels as they replicate within infected hosts; however, only a small proportion of these variants transmit between hosts and eventually expand at a global scale [20]. In this study, our results emphasized the importance of exploring virus diversity at the sub-consensus level and highlighted the significance of studying LFVs that might affect antigenic drift, enhance virulence, or reduce sensitivity to antiviral drugs. Moreover, our data signify the importance of considering within-host diversity when trying to understand variations of symptoms among patients. Finally, linking the underlying virus diversity, the existence of certain variants, and age of patients is a useful exercise that provides important insights into host-virus adaptations that might affect viral pathogenesis and evolution.

## Figures and Tables

**Figure 1 microorganisms-08-00133-f001:**
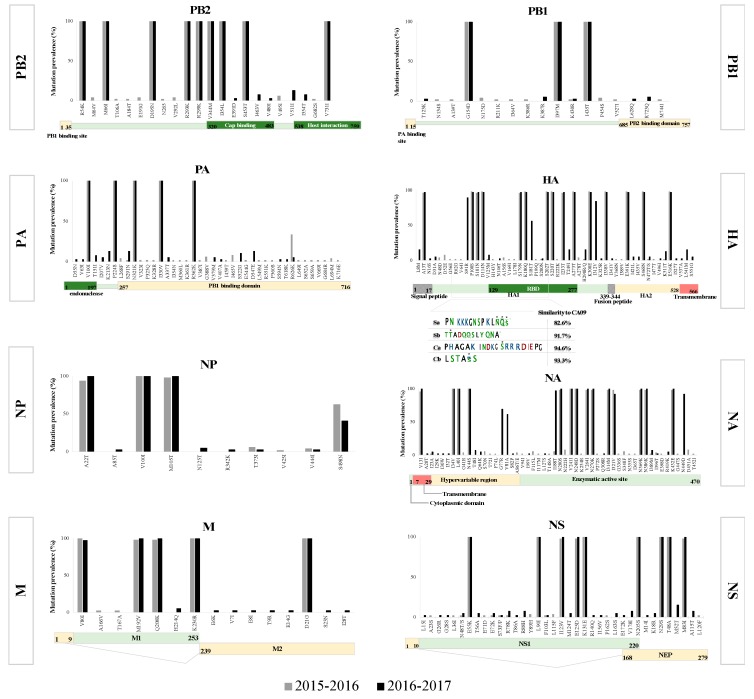
Accumulation of genetic variations in pH1N1 virus genes over two seasons (2015–2016 and 2016–2017). Prevalence of amino acid substitutions identified in each gene of 90 pH1N1 viruses collected during the flu seasons of 2015–2017. Amino acid substitutions were identified in comparison to reference virus A/California/07/2009 (EPI-ISL-227813). Schematic representations of amino acid positions with respect to gene structure are shown below the x-axis of each graph. H1 and N1 numbering were used for HA and NA genes, respectively, while numbering started from the first methionine (M) in the coding region for the rest of the genes. For the HA gene specifically, analysis of amino acid substitutions in HA antigenic sites is shown. Similarities of these sites to the respective reference virus (A/California/07/2009) were calculated using the Flusurver website (flusurver.bii.a-star.edu.sg/). Amino acids showing variations are denoted with a star.

**Figure 2 microorganisms-08-00133-f002:**
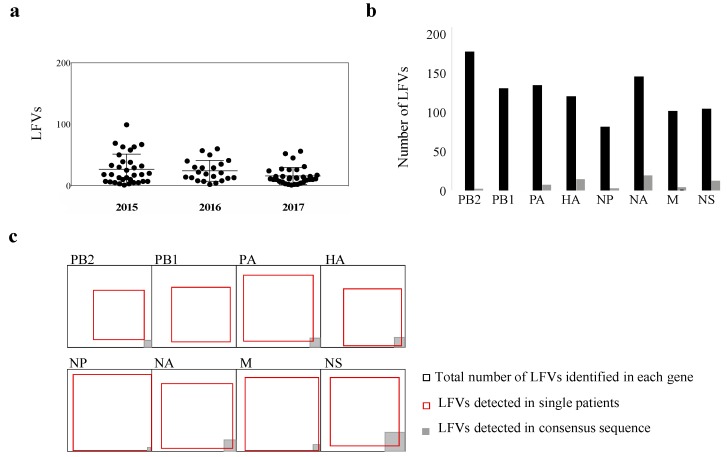
Overall representation of non-synonymous low-frequency variants (LFVs) identified across genes of pH1N1 viruses sequenced during 2015–2017. (**a**) Total number of LFVs (frequency less than 50%) identified in each patient sample. The lines indicate the mean number of LFVs in each year. (**b**) Total number of LFVs reported in each gene (black) and number of LFVs that reached the consensus level (frequency of more than 50%) (gray) among viruses sequenced in this study. (**c**) Schematic overlap representation of total number of LFVs identified in each gene (black square), LFVs observed once (red square), and LFVs observed at the consensus sequence level of HA sequences analyzed in this study (gray shaded square).

**Figure 3 microorganisms-08-00133-f003:**
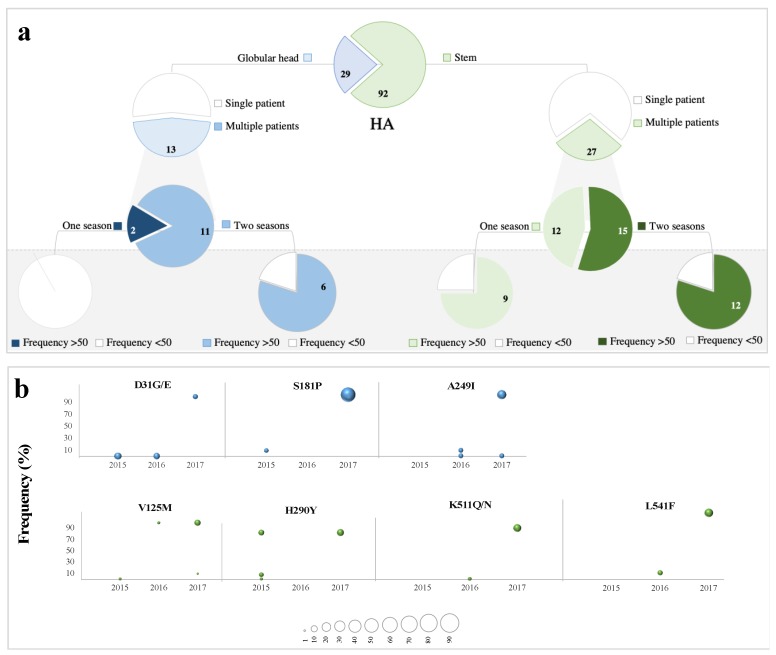
(**a**) Numbers of non-synonymous LFVs in HA sequences with respect to their genomic position (globular head and stem), prevalence (single patient and multiple patients), and temporal distribution (one-season and two-season appearances). Variants are colored based on their position in the HA gene: blue for the HA globular head, green for the HA stem. Pie charts in the shaded area represent the number of LFVs seen at the consensus sequence level: dark blue (in the globular head) and dark green (in the stem) colors represent LFVs observed only in one season but that were found in HA consensus sequences; light blue (in the globular head) and light green (in the stem) colors represent LFVs observed in both seasons and were found in HA consensus sequences. Amino acids in rectangles indicate variants seen in the consensus sequences of pH1N1 viruses both locally and in the AAE sequences. The prevalence of LFVs at the consensus level of AA sequences was determined using 13,021 HA and 10,961 NA sequences deposited in IRD and GISAID during the 2015–2018 period from Asia, Europe, and Africa. (**b**) Temporal tracking of identified non-synonymous LFVs in HA genes over a three-year period (2015–2017). Variants are colored based on their location in the globular head (blue) and stem (green) subunits. The bubble size represents the prevalence of each variant among sequenced viruses in each year (as indicated in the scale below the scale).

**Figure 4 microorganisms-08-00133-f004:**
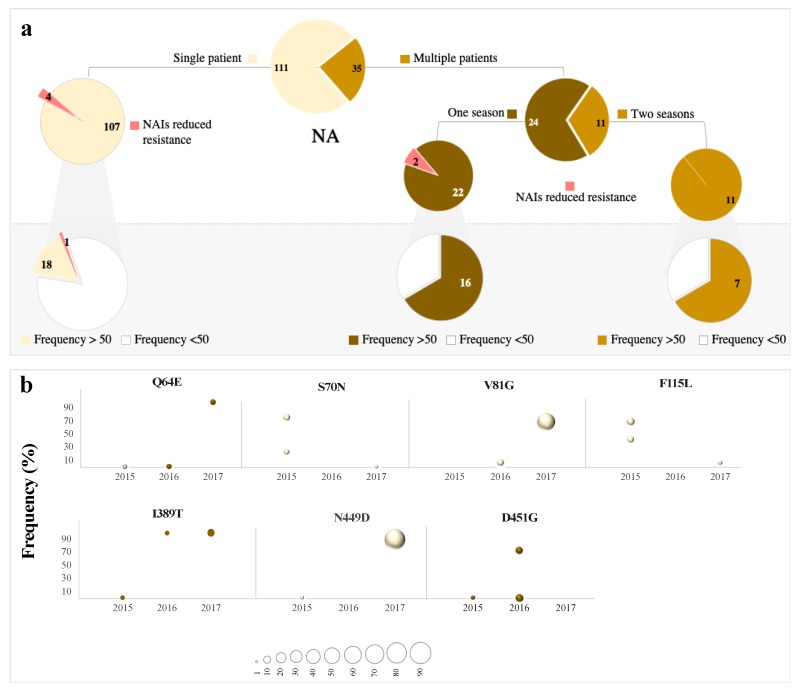
(**a**) Number of non-synonymous LFVs identified in the NA gene of pH1N1 viruses with respect to their prevalence (single patient and multiple patients) and temporal distribution (one-season and two-season appearance). Variants are colored based on their prevalence: brown for variants exhibited by two or more patients and light brown for variants that occurred once. LFVs associated with reduced NAI sensitivity are indicated in red. Pie charts in shaded areas depict LFVs seen at the consensus sequence level. Amino acids in rectangles indicate variants seen in consensus sequences of pH1N1 viruses both locally and AAE sequences. (**b**) Temporal tracking of non-synonymous LFVs in NA genes during 2015–2017. Variants are colored based on their appearance either in a single patient (light brown) or in multiple patients (dark brown). Prevalence of each variant in any specific year is represented by the bubble size (as indicated in the scale below the graph).

**Figure 5 microorganisms-08-00133-f005:**
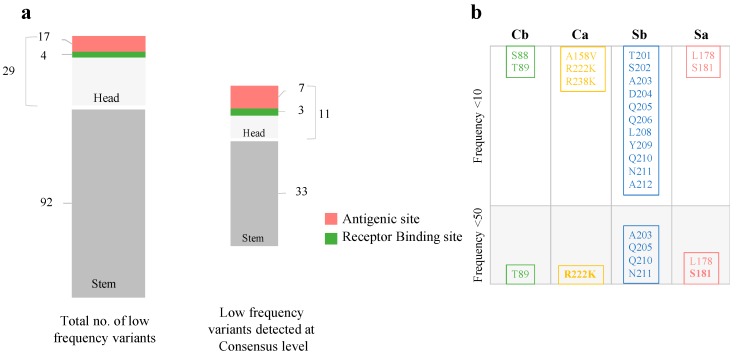
Distribution of LFVs across the HA gene (**a**) The left bar represents the total number of LFVs reported in HA subunits of 90 samples collected during 2015–2017. Variants reported in the head region were further subcategorized based on their position in RBS (green) and/or antigenic sites (red and yellow). The right bar represents the number of LFVs that appeared at the consensus sequence level of pH1N1 viruses isolated during the study period. (**b**) Bars show distributions of LFVs in HA antigenic sites. The upper panel illustrates all LFVs identified in each site while the lower panel represents the number of variants that successfully reached the consensus sequence (bold) locally and in AAE sequences.

**Figure 6 microorganisms-08-00133-f006:**
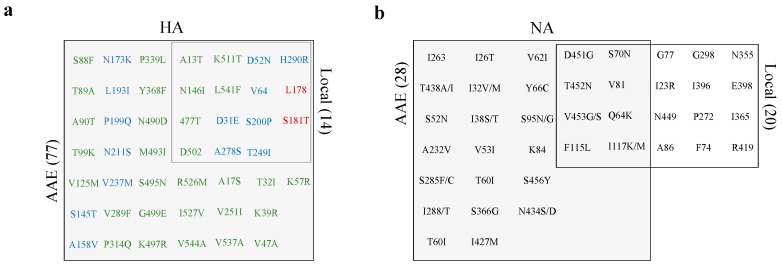
Venn-diagrams displaying LFVs observed in HA and NA consensus sequences during 2015–2017 locally and during 2015–2018 in AAE sequences. (**a**) A total number of 14 and 75 variants were detected in consensus sequences from locally and AAE sequenced viruses, respectively. Here, we only present variants detected in more than two sequences. Variants are colored with respect to their position in HA: blue for the HA head subunit, green for the HA stem, red for the Sa antigenic site (**b**) A total number of 20 and 28 variants were detected in NA consensus sequences from locally and AAE sequenced viruses, respectively.

**Figure 7 microorganisms-08-00133-f007:**
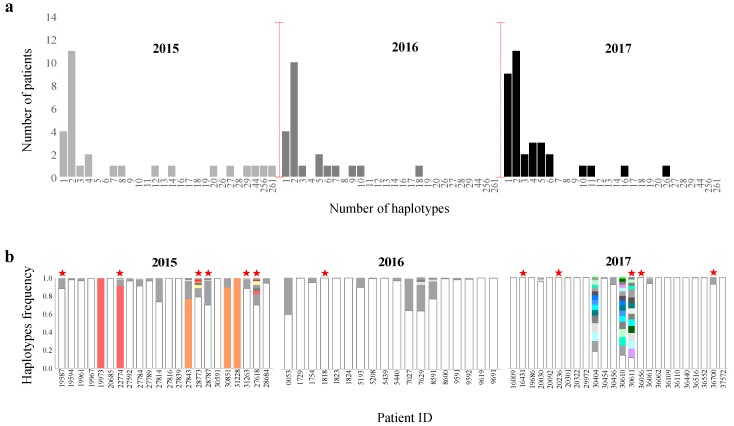
(**a**) Number of HA haplotypes identified in each patient sample. The number of haplotypes assembled from each sample was estimated using the conservative mode of haplotype reconstruction in QuasiRecomb tools. A total of 82 samples with HA coverage >1000x were included in this analysis. Haplotype reconstruction was done for the nearly full-length HA gene (18–1680 nt). (**b**) Diversity and frequency of HA haplotypes identified within patients. Total number of HA haplotypes and their frequencies were determined using the conservative mode implemented in QuasiRecomb; then, haplotypes were ranked based on their frequencies from high to low (bottom to top). Haplotypes with different genetic compositions (different by at least one nucleotide) are indicated in gray shades. Colored haplotypes denote haplotypes shared by more than one patient. Samples with severe symptoms are labeled with a red star. Only samples harboring more than 2 and less than 50 haplotypes are presented here. For simplicity, two samples (28711 and 27825) have been excluded from this figure due to the large number of haplotypes (261 and 256 haplotypes, respectively).

**Figure 8 microorganisms-08-00133-f008:**
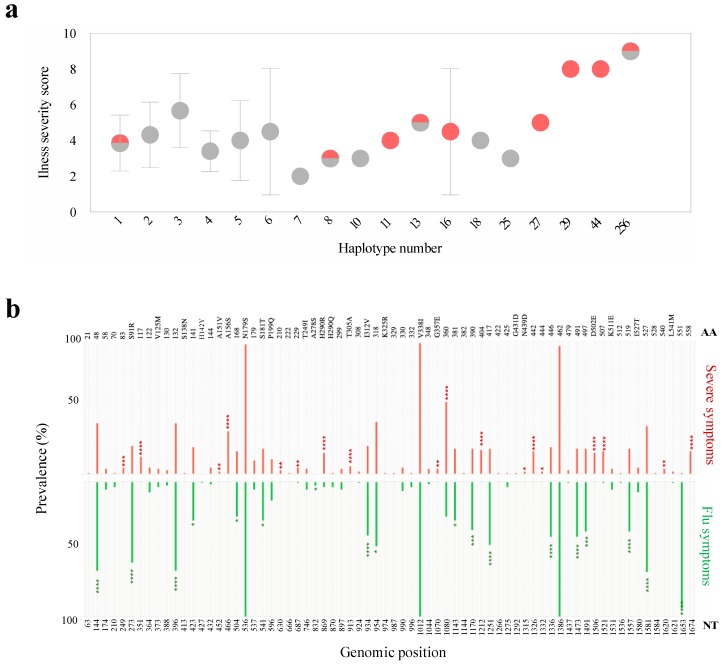
(**a**) Relationship between within-host HA diversity and illness severity. A Poisson distribution was used to examine the relationship between HA diversity and severity of illness. Illness severity was assessed as previously defined by Capelastegui et al. (2012): age >45 years, male gender, number of comorbidities (asthma, chronic pulmonary disease, cardiovascular disease), pneumonia, dyspnea, and confusion (Capelastegui et al., 2012). Each circle represents the mean severity score within groups of patients carrying the same number of haplotypes. The red color in circles denotes patients who are either over the age of 60 or under the age of 5. (**b**) Comparison of prevalence and localization of single nucleotide variants in HA haplotypes of patients with severe symptoms (red) and flu-like symptoms (green). AA at the top of the graph indicates changes at the amino acid level while NT at the bottom indicates mutations at the nucleotide level. Statistical analysis was done using the chi-square test in Prism7 to determine significance (* *p*-value < 0.01, ** *p*-value < 0.001, *** *p*-value < 0.0001 **** *p*-value < 0.00001).

**Figure 9 microorganisms-08-00133-f009:**
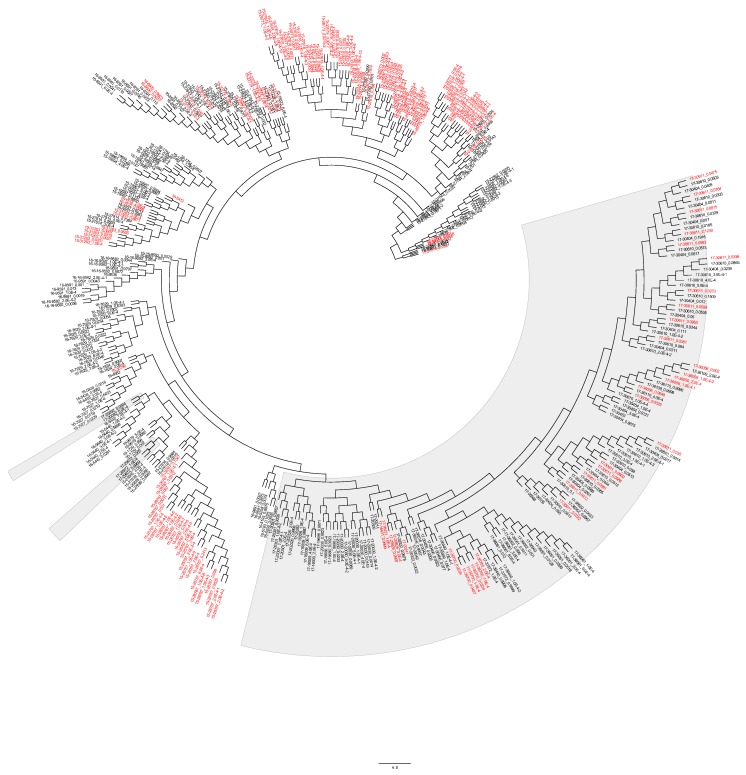
Phylogenetic relationships of 430 HA sequences representing consensus sequences and haplotypes of all samples. The HA sequence name is divided as follows: the first two digits represent the year of isolation followed by the sample number then frequency of each haplotype as determined by QuasiRecomb tools. Sequences of 2017 clustered together (gray shade) while those of 2015 and 2016 were interspersed on the opposite side of the tree. The phylogenetic tree was constructed using the neighbor-joining method with bootstrapping of 1000 replicates (CLC genomic workbench v11). Haplotypes found in patients with severe respiratory and/or cardiac manifestations are colored in red.

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
