# Peer review of "Inter-Versus Intra-Host Sequence Diversity of pH1N1 and Associated Clinical Outcomes"

_microorganisms, 2020, doi:10.3390/microorganisms8010133_

Round 1

Reviewer 1 Report

The authors make a serious effort to analyze inter- versus intra-host sequence diversity in pH1N1 samples in their region from 2015-2017. This is a short time window in a specific region and it is difficult to derive far-reaching conclusions on influenza evolution in general. The technical steps of the Bioinformatics methods are appropriate with conservative thresholds. However, the two main concerns are the use of an incomplete set of circulating sequences for comparison and the lack of statistical tests and controls to show significance of suggested associations forming major conclusions (e.g. LFV predictive for future strains, number of haplotypes higher in severe cases,...).   

In detail:

line 2/title: "inter-host and intra-population sequence diversity" mixing host and population together does not fit, maybe better "inter- versus intra-host sequence diversity"?
  line 178: "global variation": these numbers of HA and NA sequences for AAE seem a strong underrepresentation of actually available samples when looking at what is available in GISAID for example (14k in GISAID vs 500 at NCBI...)   line 347/Figure 3: Is this S181P or should it be S181T?   line 373 and line 457: "would help us predicting next season variants" one cannot conclude this from a 1-2 examples without test for significance, one way to test this would be to first list all mutations that are fixed by 2018 and then compare how many of them are found among the quasispecies in the data from 2015-2017 and do a statistical test (Fisher's exact, chi square,...) over the 2x2 contingency matrix fixed2018-yes, fixed2018-no, quasi1517-yes, quasi1517-no     line 444: would also be interesting to get a test for significance to check if positive selected sites are enriched among LFV   line 496: does the number of haplotypes found in a sample correlate with (and hence could be biased by) higher sequencing depth/quality for these samples? The latter could also be linked to higher viral titre which could also be linked to severity.    line 538 and Figure 8: there needs to be a statistical estimate if the haplotype abundance is significantly higher in severe cases to support the presented conclusions   Minor:   line 66 Typo: "would deems important"   line 164: "NCBI Influenza Virus Resource Database (www.ncbi.nlm.nih.gov/IRD)" wrong link, this should be https://www.ncbi.nlm.nih.gov/genomes/FLU/   

Reviewer 2 Report

The article entitled « Inter-host and intra-population sequence diversity of pH1N1 and associated clinical outcomes » is a very interesting analysis of pH1N1 diversity.

The paper is very well written and interesting for a better understanding of influenza virus evolution.

The paper is based on :

- a precise analysis of low-frequency variants and haplotypes obtained by deep sequencing on nasal swabs of 90 patients collected at the virology laboratory (HMC, Doha-Qatar) between 2015-2017.

- an analysis of variants at a global scale using consensus GISAID sequences for pH1N1 between 2015-2018

- some data regarding the clinical severity

One aim of the study is to compare the diversity at the « single host » scale with the diversity across years and at a global scale to better understand the factors driving the global influenza virus evolution.

Another aim of the paper is to analyse a possible link between virus diversity and clinical severity.

The interest of the paper is due to the fact that the authors performed a real precise work to interpret the observed variants regarding to the positions of the variation.

I have only a few questions or points to be corrected :

Albeit it is interesting, the text is very long and the authors should better separate their results and the discussion to avoid repetitions.Tthis could permit to shorten the text.

The main criticism is that the figures are not visible on the available pdf version !

Line 463 : The authors correlate the higher diversity to the age of the patient (< 5 or > 60 years), it would be interesting to link this observation to the patient immunity that may be deficient in these patients.

Line 470 : some 21 variants were exclusively found in haplotypes of patients with severe complications, the authors should precise whether the initial specimens were only nasal swabs or also naso-pharyngeal aspirates or broncho alveolar lavages, as differences in the cells for viral multiplication may be linked to some variations.

Reviewer 3 Report

Interesting work. There are no figures or supplementary files attached for review. That might have been due to mistakes in submission process by the authors and/or editorial process sending out the manuscript for review.

In the context, there are too much speculation from their results. (Again, their findings are interesting, though.) Too many "interestingly" and "remarkably" in the texts.

Round 2

Reviewer 1 Report

I am ok with most of the changes and replies except this one: "You are right. There are more than 32,000 H1N1 sequences in GISAID database. However, we used sequences deposited in IRD as a representative of viruses sequenced during 2015-2018 in this part of the world. Awe agree with you that analyzing this huge number of sequences would help in supporting our results but the sequences we used have also proven our point. "

It should be explicitly included in the method section what has been the justification and procedure of selecting ~500 sequences as being representative for "global variation" when there are ~14000 (AAE region, H1N1pdm 2015-2018) available in more comprehensive databases. GISAID includes systematic and representative data from all WHO regions whereas the NCBI-based databases such as IRD have a North-America bias and as shown above cover only a fraction of the global surveillance efforts. So, it is not clear why this database was used especially for this analysis focusing on Asia, Africa and Europe. If the point is to compare the low frequency variants with global variation the used dataset should indeed be representative, which is currently not the case.

Round 3

Reviewer 1 Report

The manuscript has been improved with the comprehensive frequency analysis and the previous concern has been addressed.